# Improving View Independent Rendering: Towards Robust, Practical Multiview Effects

This paper describes improvements to view independent rendering (VIR) designed to make its immediate application to soft shadows more practical, and its future application to other multiview effects such as reflections and depth of field more promising. Realtime rasterizers typically realize multiview effects by rendering a scene from multiple viewpoints, requiring multiple passes over scene geometry. VIR avoids this necessity by crafting a watertight point cloud and rendering it from multiple viewpoints in a single pass. We make VIR immediately more practical with an unbuffered implementation that avoids possible overflows, and improve its potential with more efficient sampling achieved with orthographic projection and stochastic culling. With these improvements, VIR continues to generate higher quality real time soft shadows than percentage-closer soft shadows (PCSS), in comparable time.

**ACM Reference Format:**
. 2020. Improving View Independent Rendering: Towards Robust, Practical Multiview Effects. 1, 1 (April 2020), 5 pages. https://doi.org/10.1145/nnnnnnn.nnnnnnn

## 1 INTRODUCTION

As computer graphics hardware has improved, so has its interactive imagery, moving from line drawings to filled polygons, to textured surfaces with specular reflections. However, further improvements in visual realism — effects such as soft shadows, depth of field, and object reflections — have been hindered by current hardware, which requires multiple model traversals to render the many views needed to sample area lights, different focal depths, and reflections.

View-independent rasterization (VIR) avoids the complexity of multiple rendering passes [9] by using points as a display primitive. For every frame, it carefully transforms input triangles into a point cloud specialized to the current set of views. VIR then renders these views in parallel using the point cloud, with an order of magnitude fewer passes over the geometry.

This short paper presents our contributions to VIR, designed to increase its immediate practicality and future potential:

- *Practicality:* We improve VIR to eliminate the use of a point buffer, freeing developers from the necessity of buffer management to avoid overflow.
- *Potential:* We improve VIR's orthogonal projection, making sampling more parsimonious and "watertight" (without holes). We also introduce stochastic culling of sub-pixel triangles to reduce sampling rates further.

Author's address:

We verify that our improvements do not affect quality and performance by using them to render soft shadows. When rendering shadows comparable to those produced with a traditional, high-quality multipass technique, VIR continues to produce them in nearly an order of magnitude less time. When making shadows at practical few-millisecond speeds, VIR shadows are still of higher quality than percentage-closer soft shadows (PCSS) [2]. While our improvements in sampling do not realize performance improvements for soft shadows, we anticipate that they will for other multiview effects with heavier shader loads.

## 2 RELATED WORK

Rendering realistic imagery requires accurate simulation of light flow. However, accurate sampling of the light flow integral [4] can be difficult, particularly for effects such as soft shadows, depth of field (defocus blur), motion blur, and indirect reflections [5]). With rasterization hardware, often the fastest way to produce such samples is multiview rendering: storing many off-screen views in buffers, and combining them to produce a final view. However, the high cost of multiview rendering often requires sparse sampling and filtering to reduce resulting noise [11].

To sidestep rasterization's limitations for multiview rendering, we rely on points [6]. In today's applications, triangles often outnumber pixels, leading many to argue that points are a better rendering primitive [3]. Yet points are not widely used since their discontinuity can create "holes" when views change. Existing point renderers, therefore, use dense point clouds that render slowly; or sparse clouds with complex reconstruction that again render slowly, or produce low-quality imagery.

To improve point rendering and support multiview rendering, VIR [9] exploits rasterization hardware, which efficiently transforms triangles into points. For each frame, VIR generates a cloud of points customized to the current set of views in real time. It then renders the cloud in parallel into multiple views, reducing the number of geometry passes by a factor of ten. To accomplish this, for any triangle visible in at least one view, the geometry shader computes specialized viewing and projection matrices that center the triangle, orient it parallel to the view plane, and achieve a watertight sampling rate. It then applies the matrices to the triangle and rasterizes it to generate points. Next, the fragment shader writes each point to a buffer. When all points have been generated, the compute shader passes over this buffer, transforming and projecting each point into multiple views.

## 3 IMPROVING VIEW INDEPENDENT RASTERIZATION

We improve on Marrs et al.'s VIR implementation [9] in three ways to increase its practicality and potential. First, our implementation is bufferless, ending the possibility of overflow and simplifying VIR's use in practice. Second, we improve Marrs et al.'s orthogonal projection matrix, reducing the size of the resulting point cloud. Finally, we stochastically cull sub-pixel triangles, shrinking the point cloud

further to improve speed. The resulting improved algorithm is already practical, with better soft shadows than PCSS at comparable speeds. Improved VIR also has the potential to bring both this practicality and a significant performance improvement to more complex multiview effects.

### 3.1 Bufferless VIR

Marrs et al.'s VIR implementation stored points in a buffer, which the compute shader then rendered to multiple views. The buffer size must be set before run time, making overflows possible. Instead, we use a bufferless, entirely in-pipeline scheme that eliminates the possibility of run time buffer overflows. Rather than sending each point to a buffer, the fragment shader writes the point to multiple off-screen buffers for deferred shading. While we did not directly compare the speed of our unbuffered implementation to Marrs et al.'s buffered implementation, we expect the performance to be similar, as described in a related comparison by Marrs et al. [8].

### 3.2 Watertight and Efficient Orthographic VIR

Marrs et al.'s computation of the watertight multiview sampling rate $s_{mv}$ assumed that the fields of view for the point generation, offscreen and eye views were identical. But this cannot be true when the point generation view uses orthogonal projection, and other views do not. Unfortunately, point generation in perspective leads to sampling inefficiencies, with points spread unevenly across each triangle due to perspective distortion.

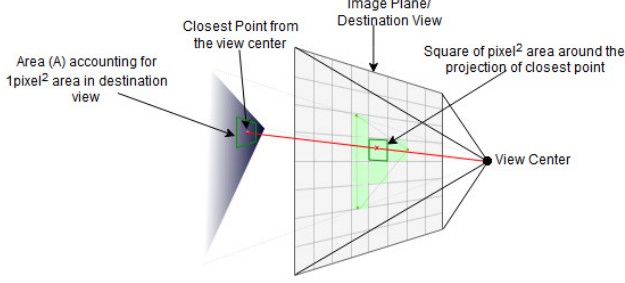

Fig. 1. Improved Orthogonal Sampling Rate.

Algorithm 1 shows the VIR algorithm, improved to use orthogonal projection. For each polygon, we find $\rho_{mv}$, the maximum point density on the projected polygon's surface, across all views as illustrated in the figure 1. For each polygon $p$, we first find the closest point on the polygon from a given view $v$ [1]. This point has the highest sample density on the polygon for that view. We compute the area of a reverse-projected pixel centered on that point $area_{P,v,p}$ by reverse-projecting its corners onto the polygon in model space. Across all views, the maximum sampling density $\rho_{mv}$ is given by the equation (1), and the orthogonal scaling factor $s_m v$ is given by (2)

$$\rho_{mv} = \forall_{v \in V} \ max(\rho_{mv}, \frac{area_p}{area_{P,v,p}}) \qquad (1)$$

$$s_{mv} = max(s_{mv}, w \times \sqrt{\frac{\rho_{ortho}}{\rho_{mv}}}) \qquad (2)$$

---

**Algorithm 1** View Independent Rasterization

   In Geometry Shader Stage:
1: **for** each polygon (P) **do**
2:   **for** each viewpoint (v) **do**
3:     $c_p$ = Closest point on the polygon from the viewpoint
4:     $area_p$ = Area of pixel
5:     $area_{P,v,p}$ = Area of reverse-projected pixel centered at $c_p$
6:     $\rho_{mv} = max(\rho_{mv}, \frac{area_p}{area_{P,v,p}})$
7:   **end for**
     Compute orthogonal scaling factor
8:   $s_{mv} = max(s_{mv}, w \times \sqrt{\frac{\rho_{ortho}}{\rho_{mv}}})$
9:   Apply VIR matrix ($T_{VIR}$) and projection matrix ($T_{ortho}$) to the polygon (P)
     $P' = T_{ortho} \times T_{VIR} \times P$
10:   Send the transformed polygon (P') to the rasterizer
11: **end for**
   In Fragment Shader Stage:
1: **for** each viewpoint (v) **do**
2:   Write the generated point into the corresponding buffer using atomic write operations.
3: **end for**

---

where $V$ is the set of all view centers of destination views, $w$ is the perspective distortion, $area_p$ is the area of the polygon in model space, and $\rho_{ortho}$ is the sampling density for VIR's orthographic projection, which depends on the chosen viewing volume.

The orthographic projection matrix is given in the equation (3).

$$T_{ortho} = \begin{bmatrix} s_{mv} & 0 & 0 & 0 \\ 0 & s_{mv} & 0 & 0 \\ 0 & 0 & \frac{2}{z_{near}-z_{far}} & \frac{z_{far}-z_{near}}{z_{near}+z_{far}} \\ 0 & 0 & 0 & 1 \end{bmatrix} \qquad (3)$$

Our new orthogonal projection technique had minimal impact on VIR's performance and image quality, with orthogonal and perspective projection (as described by Marrs et al. in [9]) producing nearly identical soft shadows at similar speed. While orthogonal projection generated fewer points than Marrs et al.'s perspective projection, it required more time to do so, particularly in models with more triangles. For example, when generating 128 views for a model with 2 million triangles, orthogonal point generation rendered $122K$ points in 18.74ms, whereas perspective sampling generated $238K$ points in 17.55 ms. However, we expect that for most other multiview effects (e.g. environment mapping), increased shading loads will make orthogonal point generation's smaller point clouds advantageous.

### 3.3 VIR with Stochastic Culling

To improve speed further, we stochastically cull (and avoid generating points for) triangles that span less than $1/8^{th}$ of a pixel in VIR's point generation view. The smaller the proportion of the pixel covered by the triangle $T_{pp}$, the more likely $T_{pp}$ will be culled, with probability 1 - ($8 \times T_{pp}$). Stochastic culling breaks our watertight guarantee, but we have not yet observed any holes in soft shadows

generated with models ranging from $30K$ to $2M$ triangles. For example, Figure 5 shows a perceptual comparison of improved VIR with and without culling using HDR-VDP2 [7]; cool heatmap colors indicate little or no difference. Across this same range of models, we found that stochastic culling is most beneficial when subpixel triangles are common. This replicates Marrs et al's [9] finding that for large triangles (spanning dozens of pixels or more), VIR is less efficient than standard rasterization.

## 4 RESULTS

We demonstrate the practicality and potential of improved VIR with soft shadows. Below, we offer comparisons to both high quality and high-speed shadow algorithms, as well as a brief comparison to Marr's et al.'s implementation [9].

### 4.1 High Quality Comparision

As an evaluation platform, we used OpenGL 4.5 on a PC with an Intel i7-8700K @ 3.70 GHz CPU and an NVIDIA 1080Ti GPU, running Windows 10 OS. We rendered several scenes, with detail concentrated in the central 30% of the field of view. All scenes were dynamic, rotating around themselves twice (720 degs), while lights remained stationary, casting moving shadows. We used 32-bit unsigned depth buffers, with a resolution of $1024^2$. For each light source sample, we set field of view to $45^o$.

For VIR, we used all three of our improvements: a bufferless implementation, orthographic point generation, and stochastic culling. Like Marrs et al., we produced 128 views in four passes (32 per pass, the warp size of our GPU). As a high quality comparison, we used multiview rendering (MVR), which used 128 passes to create 128 standard shadow maps [12]. To compare the performance of these methods, we averaged GPU run-time and the number of points generated over 1256 frames of execution.

Table 1 shows results for several models [10]. The leftmost column shows the number of triangles per model. The adjacent three show improved VIR's point cloud size, the time required to generate that point cloud, and the total time to generate VIR's point cloud and construct depth maps (with results including stochastic culling in brackets). For comparison, the next column reports the total time taken by MVR to make depth maps, and the rightmost column reports performance improvement as the ratio of MVR time over VIR time, highlighted in blue. The illumination technique is the same for VIR and MVR, we do not include it. VIR renders these dynamic, complex soft shadows up to 3.4 times faster than MVR without stochastic culling, and up to 7 times faster with it.

Figure 2 shows soft shadows generated by VIR and MVR. Though VIR is faster than MVR, its visual quality is quite similar to high quality MVR and stable under animation. VIR includes the hallmarks of high quality shadows, such as soft penumbras and contact hardening (sharper shadows closer to the light). Because VIR and MVR both use shadow mapping and differ only in how they generate depth buffers, both suffer the same artifacts (e.g. "peter panning" and acne). Note that the breaks in the dragon's shadow with VIR are smaller than in MVR; VIR's view independent samples silhouettes more densely than view dependent MVR.

**GPU Performance of VIR [with stochastic culling]**

**Soft Shadows for 128 Views**

| Models (# tris) | VIR # points | VIR pt gen (ms) | pt gen + depth (ms) | MVR (ms) | × Faster |
|---|---|---|---|---|---|
| Tree (151.7K) | 275.0K [228.9K] | 0.82 | 3.80 [2.54] | 7.33 | 1.92 [2.88] |
| Dragon (883.3K) | 684.6K [489.6K] | 12.91 | 16.58 [13.78] | 57.01 | 3.44 [4.13] |
| Buddha (1.1M) | 586.1K [225.6K] | 8.49 | 21.52 [12.22] | 69.33 | 3.22 [5.67] |
| Lucy (2.0M) | 1.1M [250.8K] | 12.91 | 38.00 [17.18] | 122.31 | 3.22 [7.12] |

Table 1. Speed comparisons of View Independent Rendering (VIR) and Multiview Rasterization (MVR), with Models in the left column, VIR in the middle three, and MVR to the right. We highlight VIR's performance improvements in blue. Results using stochastic culling in are in brackets ([]).

### 4.2 High Speed Comparison

To evaluate the use of improved VIR in a more practical, real time setting, we compare improved VIR to percentage-closer soft shadows (PCSS) [2]. We generated soft shadows using 16 views in 2.6 ms and compared it to an image generated by PCSS with 96 samples per pixel (32 blockers and 64 filter samples), in 2.5 msec. The resulting images are shown in Figure 3. Figure 4 shows the perceptual comparison of these images against a reference 128-view MVR solution using HDR-VDP2 [7]. The image generated by VIR has less error than PCSS, especially at the region where the rods cast shadows on the dragon.

To further gauge the effect of the size of the triangles on our improved VIR technique, we sorted triangles into three classes: *subpixel triangles*, with the length of the longest side less than a pixel; *supra-pixel 1 triangles*, with the longest side length greater than a pixel and less than 10 pixels; and *supra-pixel 2 triangles*, with longest side of the triangle is greater than 10 pixels. We then studied our technique on the scene which had mostly subpixel triangles, mostly supra-pixel 1 triangle and mostly supra-pixel 2 triangles. We observed that with the majority of triangles being sub-pixel, we were able to achieve a huge speedup. With a large number of supra-pixel 1 polygons, VIR was still faster than MVR. But with supra-pixel 2 polygons, VIR didn't do well and MVR proved to be a better candidate. One way to handle a scene with a mix of all sizes of polygons is to have a hybrid of VIR and MVR method. By rendering small polygons with VIR pass and the large polygons with MVR pass.

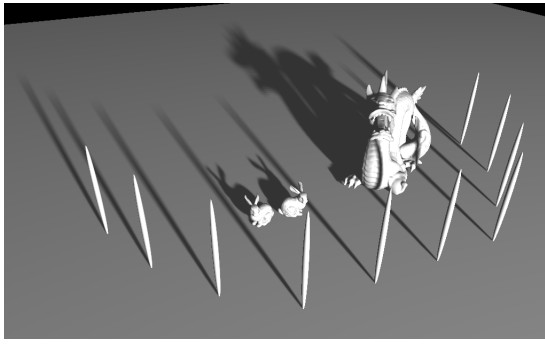 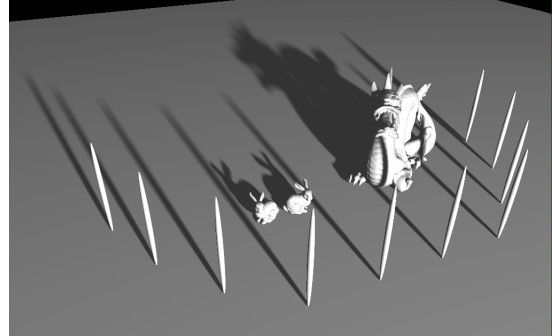

Fig. 2. Soft shadows generated using MVR (left) and using our VIR implementation (right). Both generate 128 depth maps. MVR takes 57 ms to generate them, whereas our implementation generates points and depth maps in 16.6$ms$.

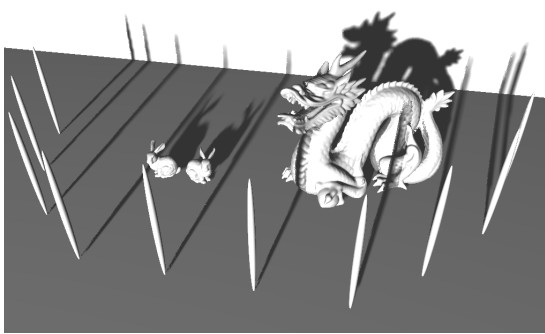 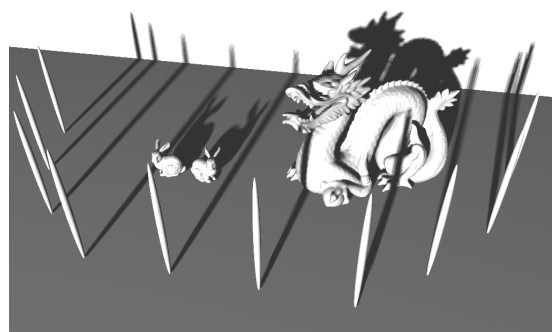

Fig. 3. (Left) Shadow rendered using PCSS (32 blocker and 64 PCF samples). Shadow artifacts can be seen on the dragon, where 2 rods cast shadows on it. (Right) Soft shadows rendered with our VIR implementation using 16 views and all improvements. PCSS takes 2.5 ms, whereas our implementation 2.6 ms. delivering better quality.

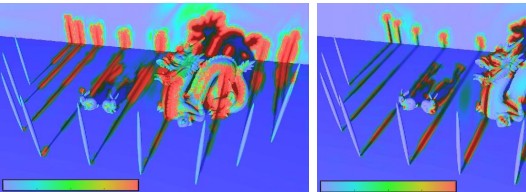

Fig. 4. A perceptual comparison of Figure 3's images. HDR-VDP2 compares each to a 128-view MVR render of the same scene. Red indicates a more perceivable difference.

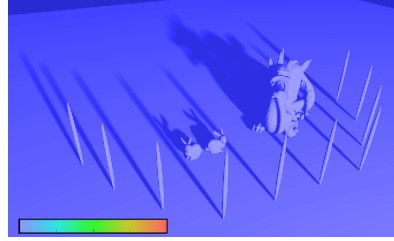

Fig. 5. A perceptual comparison of improved VIR with and without stochastic culling. HDR-VDP2 shows little or no difference.

### 4.3 Improvements Comparison

Our VIR improvements did not improve soft shadow quality or speed over Marrs et al.'s implementation. We expect our improvements to show their merit for other, more demanding multiview effects, such as environment mapping and defocus blur.

We did not directly compare improved VIR to the original on the same hardware, but Marrs et al.'s results are similar to our own, on comparable hardware. Although improved VIR did generate sparser — but still watertight — point clouds, the effort required to do so canceled out performance gains for soft shadowing. However, soft shadowing shader loads are minimal: only a depth comparison is required. Other multiview effects require much more complex shaders and should realize the performance benefits of our VIR improvements, making those effects practical as well.

## 5 LIMITATIONS, CONCLUSIONS AND FUTURE WORK

Marrs et al.'s original VIR implementation was able to cull points in the compute shader by comparing several local points and rendering only the closest (unoccluded) sample for each view pixel. Our bufferless implementation does not use compute shaders, and cannot perform this local point culling. More significantly, our stochastic triangle culling breaks the guarantee of watertight sampling, though it has not created holes in our testing.

Despite these limitations, the VIR improvements we describe here make VIR immediately more practical and promise wider utility in the future. With a bufferless implementation, developers need no longer risk runtime overflows. We also show results demonstrating higher quality soft shadows than PCSS at practical rendering speeds. In the future, we plan to explore the potential of improved VIR with multiview effects having more demanding shading loads, such as environment mapping, diffuse global illumination and defocus or motion blur. We will also study global probabilistic limits for holes resulting from stochastic triangle culling. Finally, we plan to examine applications of improved VIR to light field displays, which demand tens or hundreds of views in every frame.

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
