# OpenReview forum: "Improving View Independent Rendering: Towards Robust, Practical Multiview Effects"
_graphicsinterface.org/Graphics_Interface/2020/Conference — Submitted to GI 2020_

### Official Review · AnonReviewer2 · 2020-04-20
**Three heuristic techniques are presented to improve the rendering performance of the View Independent Rendering presented in Ref [9] in this manuscript. The results don’t show much improvement in speed or in rendered images. They expect improvement over [9] in more demanding rendering requirements (not implemented yet), than simply soft shadows (implemented). Overall there are some practical improvements, but I think with limited technical contribution.**

**Rating:** 5
**Confidence:** 3

**Review:**

The paper presents three heuristic techniques to improve the rendering performance of the View Independent Rendering presented previously in reference [9] in this manuscript.  These techniques include:
(i) A practical suggestion of not requiring to buffer the point cloud, instead send it directly to desired off-screen buffers. This would avoid buffer overflow, but I expect this could make it slower. I would have liked to see some empirical evidence that it does not reduce speed. Table 1 shows some time improvements, but the larger gain seems to be due to stochastic culling of sub-pixel size triangles (technique iii).
(ii) The second technique is to use orthographic sampling over perspective sampling used in the original paper. This eliminates non-uniform sampling due to the perspective, and yields smaller point clouds. But again I suspect it will take longer as the model size increases. It would be good to see how this affects speed as the model size increases.
(iii) The third technique is stochastic culling of small triangles which span less than 1/8𝑡ℎ of a pixel . This is a heuristic which certainly speeds up the method, but has the danger of leaving holes. Their claim is that in their example renderings it did not show holes! This is not convincing. Also, how was the factor 1/8 chosen? Why not use, say, 1/4 or 1/32? If this was chosen based on empirical studies, then it would be good to have presented them.

The results do not seem to show much improvement either in speed or in rendered image quality.  The paper claims that they expect improvement over [9] in renderings requiring more demanding shading loads, such as environment mapping, diffuse global illumination and defocus or motion blur. However this is not implemented and not demonstrated in any way. Overall I find the technical contribution rather limited, though there are some practical improvement techniques.

This paper could fit into a short contribution category, if there is one.

---

### Official Review · AnonReviewer1 · 2020-04-20
**Promising extensions to view independent rendering that require more work**

**Rating:** 5
**Confidence:** 4

**Review:**

The work proposes adjustments to an existing algorithm. One of the contributions seems to be based on a misunderstanding of the original algorithm. The other two are rather minor adjustments. The paper is short and the text states that it was submitted as a "short paper". This category does not seem to exist though. For a full paper, the work does not seem ready for publication.


This paper presents a modification of the existing work by Marrs et al., which converts a mesh into a point cloud that can then be used for various applications (soft shadows, depth of field...). The conversion process is fast by relying on the rasterization capabilities of the graphics cards. In principle, each triangle is rendered at a suitable resolution and the resulting fragments are interpreted as point primitives.

The paper claims to make three contributions.

1) The storage of the points in a buffer is avoided and the points are instead directly rendered into additional views.

2) The authors claim to correct a mistake in the original work that leads to incorrect sampling precision.

3) A culling algorithm is proposed to stochastically ignore small triangles.


The first and third contribution are small modifications. The second would be of interest but it seems that the original algorithm was potentially misunderstood.

The submission mentions that Marrs et al. setup an orthographic matrix and therefore the sampling rate cannot be correct. Nevertheless, this is only the first step. Marrs et al. state after the ortho matrix was applied:
"Next, we apply a default view-projection transformation to each polygon that positions the camera at the world origin looking down the positive z-axis."
Their first ortho matrix rotates the triangle such that its normal aligns with the z-axis and shifts it along this z-Axis according to the wanted scale factor, which is determined involving the various views. Next, the default perspective camera is applied along the z-Axis, which therefore produces the appropriate amount of pixels. This mechanism is also illustrated in Fig. 1 of their paper. With this observation, the actual contribution of the submission is reduced significantly, as the other two elements are not sufficient to grant acceptance.

Additionally, the submission would benefit from a careful rewrite. The paper assumes the knowledge of Marrs et al.'s work, which makes it less stand-alone. Furthermore, the mathematical formulas are not entirely sound.
Equation 1 is a recursive definition and also contains a forall operator that is misplaced. The initial value is left out.
The same holds for Equation 2. The pseudocode hints at what was meant but it unnecessarily complicates the reading flow.

A very positive element in this work is the evaluation. Effort was spent on implementing the soft-shadow application. The resulting shadows look convincing and even a comparison to an existing competing solution is included. Unfortunately, the current application is not performing better than Marrs et al.'s work.

Another interesting evaluation concerns the influence of triangle sizes on the performance. The paper distinguishes several cases and shows the high efficiency when sub-pixel triangles occur - where the culling mechanism has its opportunity to shine. It would have been good to show the limitations of the culling step. In principle, with a plane that is tessellated, there should be visible gaps for higher resolution levels.


Overall, it is clear that quite some work went into the submission but given the above flaws, the paper is not ready for publication. Still, the work is a nice starting point. The authors could address the above issues, add the promising new applications that they give an outlook on, and resubmit in the future.

---

### Official Review · AnonReviewer3 · 2020-04-21
**Lacks fair comparison to prior art**

**Rating:** 4
**Confidence:** 3

**Review:**

Summary: This paper suggests two improvements to the View-independent rasterization (VIR) algorithm of Marrs et al. [9]. First, it eliminates the point buffer to avoid buffer overflow issues, and second, it replaces the projection matrix utilized in the original method with an orthographic projection computed per-polygon. Further, a stochastic sampling technique is introduced to speed up the algorithm.

Overall, the paper reads well and problems are stated clearly. However, I feel that the paper lacks comparisons which would really show the impact of the proposed improvements to VIR. Comparisons are performed, but only against other methods such as PCSS and MVR. However, comparisons against the original version of VIR are missing.

In my opinion, the comparisons provided only show how VIR compares to other methods, irrespective of the improvements suggested here. Runtime comparisons against the original VIR will show that the unbuffered implementation is faster than [9] (or just as fast as it). Visual comparisons against [9] will show that the orthographic projection improves visual quality.

> Currently, the paper only offers the conjecture that these improvements will show up for more demanding effects.
> With ref. to the buffer-free implementation, it's not clear to me how the presented implementation is different from the alternative proposed by Marrs et al. [9, Figure 4 and Section 4]. They further mention that the chose the buffered technique with compute shaders since it was observed to be faster.
> I like Fig. 5 which shows that the stochastic sampling provides performance benefits without a perceptible visual degradation.

To summarize, I think this paper needs an ablation study. I would like to see comparisons of unbuffered vs. original VIR, then a visual comparisons b/w [9]'s perspective projection vs. the proposed per-polygon orthographic projection. It would also be good to include more result figures.

Some possible typos:
> Eq. 1 should be
$$\rho_{mv} = \max_{v\in V} \frac{area_p}{area_{P,v,p}$$
> Eq. 2 has the same issue.
> $\rho_{mv}$ and $s_{mv}$ are not initialized in Algorithm 1.
> The use of the term "watertight" is confusing. I would associate watertight with topologically structured data such as triangle meshes, and not with point clouds.

---

### Meta-Review · Area_Chair1 · 2020-04-23

**Recommendation:** Reject
**Confidence:** 4

**Metareview:**

The paper presents three extensions to an earlier method by Mars et al. All of them are not very significant, and do not seem to show much improvement over the earlier work in terms of quality of results, as shown in the examples in the paper. The authors are recommended to work on the anticipated improvements over Marrs et al's work in renderings requiring more demanding shading loads, such as environment mapping, diffuse global illumination and defocus or motion blur.  That could help make a case for these extensions.

---

### Decision · Program_Chairs · 2020-04-25

Reject